# Impact of Nanoparticle-Based TiO_2_ Surfaces on Norovirus Capsids and Genome Integrity

**DOI:** 10.3390/foods13101527

**Published:** 2024-05-14

**Authors:** Philippe Raymond, François St-Germain, Sylvianne Paul, Denise Chabot, Louise Deschênes

**Affiliations:** 1Canadian Food Inspection Agency (CFIA), St-Hyacinthe Laboratory—Food Virology National Reference Centre, St-Hyacinthe, QC J2S 8E3, Canada; 2Agriculture and Agri-Food Canada (AAFC), St-Hyacinthe Food Research and Development Centre, 3600 Casavant W, St-Hyacinthe, QC J2S 8E3, Canada; 3Agriculture and Agri-Food Canada (AAFC), Ottawa Food Research and Development Centre, 960 Carling Ave, Ottawa, ON K1A 0C6, Canada

**Keywords:** norovirus, TiO_2_, nanoparticles, PtCl_4_, RNA extraction, capsid integrity

## Abstract

Human noroviruses (HuNoVs) are among the main causes of acute gastroenteritis worldwide. HuNoVs can survive for several days up to weeks at room temperature in the environment, on food, and on food handling and processing surfaces. As a result, this could lead to viral spread through the ingestion of food in contact with contaminated surfaces. The development of stable surface materials with antiviral activity might be useful to reduce viral outbreaks. Metal-based compounds, including photoactivated titanium nanoparticles (TiO_2_ NPs), are known for their antiviral activity. In this study, we tested the impact of 2000 µg/mL TiO_2_ NPs, with or without UV activation, on HuNoV GII and murine norovirus. Their recovery rates were reduced by 99.6%. We also evaluated a new TiO_2_ NP-coating process on a polystyrene surface. This process provided a homogenous coated surface with TiO_2_ NPs ranging between 5 nm and 15 nm. Without photoactivation, this TiO_2_ NP-coated polystyrene surface reduced the recovery rates of intact HuNoV GII by more than 94%. When a capsid integrity treatment with PtCl_4_ or a longer reverse transcription polymerase chain detection approach was used to evaluate virus integrity following contact with the TiO_2_ NP-coated polystyrene, the HuNoV GII recovery yield reduction varied between 97 and 100%. These results support the hypothesis that TiO_2_ NP-coated surfaces have the potential to prevent viral transmission associated with contaminated food surfaces.

## 1. Introduction

Human noroviruses (HuNoVs) and other human enteric viruses, including hepatitis A virus (HAV) and rotaviruses (RoV), are considered the leading cause of foodborne illnesses and outbreaks [1,2,3]. In 2016, the global economic burden of HuNoVs was estimated at 4.2 billion in direct health system costs and USD 60.3 billion in societal costs per year [4]. In the U.S., there was an estimated 20 million annual cases, with less than 1% associated with reported outbreaks [4]. These HuNoV infections represent an annual healthcare cost of USD 430 to USD 740 million in the U.S. [5]. HuNoVs are shed in the vomit and faeces of people who are infected, including asymptomatic carriers [6]. HuNoVs can remain infectious for several days to weeks at room temperature in the environment and on stainless steel, Formica^®^ ceramic, polyethylene, and polyvinyl chloride [7,8,9,10,11,12,13,14]. This leads to the spread of the virus either directly through contaminated hands and surfaces or indirectly through the ingestion of contaminated food and water [2]. This, along with the increased awareness associated with reducing COVID-19 propagation, has generated a renewed interest in the identification of antiviral surface coatings [15,16].

Noroviruses are small (27–40 nm), non-enveloped, single-stranded RNA viruses that belong to a genetically diverse group of viruses of the *Caliciviridae* family [17]. Noroviruses are divided into 10 distinct norovirus genetic groups, which are undergoing constant evolution [18]. While norovirus genogroups I, II, IV, VIII, and IX infect humans, genogroup II (GII) is the most prevalent in humans [17]. According to Teunis et al., the HuNoV 50% human infectious dose is very low (<100 genomic equivalents (gEq)) in secretor-positive subjects [19]. There is currently no culture method established to confirm the presence of infectious HuNoV at the levels found on surfaces, in water, or on food commodities. Current human intestinal enteroïd (HIE) culture systems require inoculum equivalent to 10^5^ gEq [20]. The detection of HuNoVs relies on viral recovery from the contaminated matrix, extraction of RNA, and reverse transcription polymerase chain reaction (RT-PCR) amplification methodologies [3]. However, RNA detection does not always correlate with the virus’s integrity and infectivity, as non-infectious and fragmented RNA could also be detected [21,22]. Viability RT-qPCRs, such as capsid integrity treatments based on PtCl_4_ and genomic integrity tests based on the amplification of a long RNA strand, have been used to provide additional information on virus integrity [23,24,25]. Consequently, virus integrity measurements could provide a more accurate estimation of surface antiviral activity than standard RT-qPCRs.

Metal-based compounds, such as silver, zinc, iron, copper, and titanium, are among the most commonly studied materials as novel antiviral treatments [16]. Their virucidal activity as metal nanoparticles (NPs) could be significantly increased by their high specific surface area and high surface reactivity. These characteristics lead to higher levels of reactive oxygen species (ROS) and ROS-induced oxidative stress responsible for nanotoxicity [26]. The main virucidal mechanisms observed with metal nanoparticles include the binding or disruption of viral surface structures to prevent receptor binding; the production of metal ions and ROS that degrade the viral protein capsid and nucleic acids; the direct interaction with viral surfaces, proteins, and genetic material to damage viral integrity and inhibit protein synthesis and genome replication; and the cleavage of disulfide bonds to denature viral glycoproteins (reviewed in [27]). However some of these compounds, including copper and zinc oxide NPs, are highly cytotoxic when consumed, which limits their potential application as food contact surfaces [28].

Although there is some current debate on its safety as a food additive, the low toxicity of titanium oxide (TiO_2_) presents a major advantage for a food surface contact application. Titanium oxide nanoparticles (TiO_2_ NPs) were reported to have potential virucidal activity against noroviruses, mainly via a photocatalytic reaction [29,30]. TiO_2_ occurs as a rutile, anatase, and, more rarely, brookite crystalline structure with different photocatalytic activities and applications (reviewed in [31]). TiO_2_ NPs (<100 nm) are commonly used as food additives (E 171) in candies, salad dressing, creamers, icing, and marshmallows. Approximatively 40% of this additive is found in the nanosize range [32]. A safety margin of 2.25 mg of TiO_2_ NPs per kilogram of body weight per day was established by the European Food Safety Authority prior to the year 2021 [33]. However, due mainly to uncertainties concerning the genotoxicity of TiO_2_ nanoparticles, an EU expert panel recently concluded that TiO_2_ could no longer be considered safe as a food additive [34]. On the other hand, following an extensive review, Health Canada’s Food Directorate concluded that there is no scientific evidence to support that the food additive TiO_2_ is a concern for human health [35]. Food-grade TiO_2_ is still considered safe by the WHO and several other countries [36]. In the U.S., the typical exposure for an adult is around 1 mg of titanium per kilogram of body weight per day [37].

In this paper, we explored the potential of TiO_2_ NPs to degrade HuNoV GII using capsid and genomic integrity approaches to evaluate its impact on intact HuNoV GII. The potential antiviral activity of TiO_2_ NP-coated surfaces was tested as well, and a new approach to preparing TiO_2_ NP-coated polystyrene surfaces was developed for this purpose. The development of stable surface materials integrating TiO_2_ NPs might be a safe and useful way to prevent viral transmission associated with contaminated food surfaces.

## 2. Materials and Methods

### 2.1. Virus Stocks

Murine norovirus-1 (MNV) was provided by Dr. H. Virgin from Washington University (St. Louis, MO, USA). Aliquots of HuNoV GII.4 (CFIA-FVR-020) were obtained as stool specimens from the British Columbia Centre for Disease Control (Vancouver, BC, Canada). A 10% suspension of HuNoV faeces was prepared as previously described [38]. To prepare the virus inoculum, MNV and HuNoV were mixed in phosphate-buffered saline (PBS), pH 7.4 (ThermoFisher, Asheville, NC, USA). The HuNoV faeces suspension was diluted 1/50 and 1/10 in the inoculum used for the TiO_2_ NP suspension and TiO_2_ NP-coated surface experiments, respectively. For each assay, the inoculum viral concentrations, in genomic equivalent per microliters (gEq/µL), were assessed by RT-qPCR, as described below.

### 2.2. Nanoparticle Preparations

Small, anatase TiO_2_ NP aqueous stock dispersions (US Research Nanomaterial Inc., Houston, TX, USA) were selected based on their high specific surface areas and supplied with the characteristics described in Table 1.

Prior to the experiment (24–48 h), the stock dispersion was vortexed for 30 s and sonicated for 15 min in an ultrasonic bath (Branson 5510, Branson Ultrasonics, Danbury, CT, USA). Then, the suspension was diluted in PBS, with the pH adjusted to 7.8 using NaOH or HCl, and sonicated again for 15 min. Before each test, the diluted suspension was sonicated for 5 min, mixed vigorously, sonicated again for 10 min, and mixed.

### 2.3. Plate Surface Preparation

Corning Costar™, flat-bottom polystyrene, 12 and 24 not-treated multiple-well plates (Sigma-Aldrich, Oakville, ON, Canada) were filled with anhydrous ethanol USP (Greenfield, Tiverton, ON, Canada), covered, and incubated for 24 h at room temperature. Meanwhile, the anatase TiO_2_ NP stock dispersion was diluted in ethanol to 25 mg/mL and sonicated for 15 min. The plate wells were emptied, and 50 µL of the TiO_2_ NP-diluted dispersion was added to each well to cover the bottom (622 µg/cm^2^). The plate was incubated for 4 h at room temperature, then placed under vacuum for 4 h at room temperature (Vacuum Oven model #1430, Sheldon MFG Inc., Cornelius, OR, USA), followed by an additional 16 h at 80 °C under vacuum, and then maintained under vacuum at room temperature for a minimum of 8 h before being used.

### 2.4. Atomic Force Microscopy (AFM)

The morphology and topography of the surface of the bottom of the microplate wells were obtained using a Multimode 8 (Bruker Nano Surfaces, Santa Barbara, CA, USA) equipped with a J-type scanner operating in the PeakForce QNM^TM^ mode. Bruker FMV probes with a nominal resonant frequency of 75 kHz were used with a scan speed of 0.5–0.8 Hz.

### 2.5. Scanning Electron Microscopy (SEM)

Surface samples from the bottom of the TiO_2_ NP-coated plate wells were collected with a razor blade and mounted on aluminium stubs using a double sticky carbon tab. They were cleaned for a duration of 5 min using UV in a ZONESEMII sample cleaner (Hitachi High-Tech Canada, Inc., Toronto, ON, Canada). SEM images were captured using a Hitachi SU7000 (Hitachi, Tokyo, Japan) field emission scanning electron microscope in the variable pressure condition mode from 50 to 100 Pa. The working distance was 6 mm, at 20 kV, and the spot size was set at 3. The secondary electron and backscattered electron signals were collected using an ultra-variable detector (UVD) and a photo diode-type backscattered electron detector (PD-BSED). An image representation was also made using both the UVD and PD1 detectors.

### 2.6. Inductively Coupled Plasma (ICP)

To recover the TiO_2_ for the ICP analysis, 2 mL of deionised water was added to each of the TiO_2_ NP-coated polystyrene plate wells, and the plate were sonicated for 15 min on the surface of a Branson 5510 sonicator (Bransonic, Danbury, CT, USA). The extracts were collected in digestion quartz tubes, which were calibrated at 50 mL and sealed with safety pressure caps (SCP Science, Montréal, QC, Canada). The 2 mL extraction was repeated, following which the extracts were combined and dry-evaporated at room temperature in a vacuum oven. A 5 mL mix of sulfuric and nitric acid (4:1) (Trace metal grade, Fisher Scientific, Fair Lawn, NJ, USA) was added to each tube, and the extracts were digested in a NovaWAVE SA microwave digestion system (SCP Science, Montréal, QC, Canada). The temperature was raised to 220 °C for 15 min, maintained for 60 min, and cooled to room temperature for 15 min. The cooled extracts were diluted in 50 mL of deionised water and filtered on a 0.22 µm PVDF membrane (Chromatographic Specialties Inc., Brockville, ON, Canada). The digested sample analysis was carried out using an ICP OES Prodigy (Teledyne Leeman Labs, Hudson, NH, USA), with the peristaltic pump set at 1.4 mL/min. Tests were performed at a wavelength of 336.122 nm using a plasma radial view and integrated for 10 s over 3 replicas. The sample uptake delay time was set at 60 s.

### 2.7. Norovirus Treatment

#### 2.7.1. Virus Treatment with TiO_2_ NPs

In separate and removable Corning Costar™ 12- and 24-well plates (Sigma-Aldrich), 100 µL of PBS or diluted TiO_2_ NP suspension was added. An equivalent volume of PBS or virus preparation was added to these same wells, and this mixture was incubated for 60 min at room temperature in a Thermo Scientific 1300 Series A2 biological safety cabinet (Fisher Scientific, Nepean, ON, Canada). Selected well strips were treated with UVA (UVP^TM^ UVG-54) or UVC (UVP^TM^ UVL-56) (Fisher Scientific) or remained untreated for 5 min. The UV intensity was controlled using the Solarmeter Model 4.0 for UVA and Model 8.0 for UVC (Solar Light Company Inc., Glenside, PA, USA). The UVA intensity was between 2.0 and 2.2 mW/cm^2^ (500 and 550 mJ/cm^2^), while the UVC intensity was between 710 and 770 µw/cm^2^ (177 and 195 mJ/cm^2^).

#### 2.7.2. Virus Treatment with TiO_2_ NP Surface Plate

In each well of the TiO_2_ NP-coated surface plate or control plate, 10 µL of virus preparation were added to 200 µL of RNase-free phosphate buffered saline (PBS) (Wisent, St-Bruno, ON, Canada). The plates were incubated for 2 h in a biological safety cabinet at room temperature without cover.

### 2.8. Capsid Integrity Treatment

To evaluate the integrity of HuNoV GII.4 and MNV capsids, the treated viruses were incubated for 10 min with 2.5 mM PtCl_4_ (Sigma-Aldrich) at 4 °C and 300 rpm on a Thermomixer C (Eppendorf, Mississauga, ON, Canada). Control-treated virus samples without PtCl_4_ were processed in parallel. A total of 2 µL of 500 mM EDTA (Sigma-Aldrich), 5 µL of RNA carrier (Qiagen, Mississauga, ON, Canada) and 2 µL of 1% Tween 20 (Sigma-Aldrich) were added to samples with or without PtCl_4_ treatment prior to the RNA extraction.

### 2.9. RNA Extraction

A total of 500 µL of RNeasy lysis buffer RLT (Qiagen) and 0.02 M Dithiothreitol (DTT) (ThermoFisher) were added to the treated virus with or without PtCl_4_ by pipetting up and down. The lysed virus was transferred to a RNeasy column and extracted using the QIAcube supplemented with DNase I, as previously described by the manufacturer (Qiagen). The purified RNA was eluted in 50 µL of RNase-free water, and 40 units of RNasin Plus RNase Inhibitor (Promega, Madison, WI, USA) was added before its storage at −80 °C.

### 2.10. Short RT-qPCR

Both MNV and HuNoV GII.4 were quantified by RT-qPCR using the TaqMan Fast Virus 1-Step Master Mix (ThermoFisher), as described previously [38]. Briefly, the primer and probe sets developed by Baert et al. [39], Kageyama et al. [40], and Loisy et al. [41] targeting the ORF1 and ORF2 junction regions were used to generate HuNoV and MNV amplicons of 86 bp and 108 bp, respectively.

### 2.11. Long-Range RNA Quantification (Long RT-qPCR)

A long-range, two-step reverse transcription (RT) process, followed by a qPCR detection protocol, was performed, as described previously [25]. Briefly, 2.5 kb of complementary DNA (cDNA) was synthesized using the Tx30SxN primer and Maxima H Minus reverse transcriptase (ThermoFisher). A real-time PCR (qPCR) was performed on a QuantStudio 6 (ThermoFisher) using the Taq Platinum PCR kit and the primers listed in the short RT-qPCR section, following the manufacturer’s recommendations (ThermoFisher).

### 2.12. Recovery Yields

The impact of the TiO_2_ NP suspension and coated surfaces on the norovirus recovery yields were estimated using the cycle threshold (Ct) variation compared to the inoculum reference level either by RT-qPCR or long RT-qPCR. The virus recovery yield = 10^(Δ*Ct/m*)^ × 100%, where Δ*Ct* = *Ct_treated_* − *Ct_inoculum_* is the *Ct_treated_* value of extracted viral RNA from the treated samples minus the *Ct_inoculum_* value of viral RNA extracted from the inoculum, and *m* is the slope of the virus RNA transcript standard curve, or the virus dilution, for the RT-qPCR, or the long RT-qPCR, respectively.

### 2.13. Statistical Analysis

The non-parametric Friedman test, with a significance level of *p* < 0.05, was used to identify statistically significant differences in terms of recovery yields following exposition to TiO_2_ NPs or TiO_2_ NP-coated surfaces using MedCalc (v 19.3.1) (MedCalc Software Ltd., Ostend, Belgium). The Conover test (*p* < 0.05) was used as a post-hoc test to perform sub-group pairwise comparisons. Outlier values were identified and omitted using the Tukey procedure using the MedCalc application.

## 3. Results

### 3.1. TiO_2_ Nanoparticle Antiviral Activity and UV Impact in PBS

We first explored the antiviral activity of TiO_2_ NPs in suspension (Figure 1). The average HuNoV and MNV inocula were 4.4 ± 0.3 × 10^4^ gEq and 2.3 ± 0.3 × 10^6^ gEq, respectively. The HuNoV and MNV RT-qPCR efficiency ranged from 0.94 to 1.00, while the long RT-qPCR efficiency ranged from 0.9 to 0.92. Without the UV treatment, the PtCl_4_ capsid integrity treatment reduced the HuNoV recovery rate to 47 ± 2% (*n* = 15). When HuNoV GII were incubated with 20 µg/mL of TiO_2_ NPs, the viral recovery rate was reduced to 30 ± 2% (*n* = 12) with or without the PtCl_4_ treatment (*p* < 0.05). When the concentration of TiO_2_ NPs was increased to 2000 µg/mL, the HuNoV recovery rates, with or without the PtCl_4_ treatment, were further decreased to 0.3 ± 3% (*n =* 6) and 0.2 ± 3% (*n =* 6), respectively (*p* < 0.05). A slight increase in recovery rate at 108 ± 2% (*n =* 9) was noticed when the HuNoV was treated with UVA alone. This recovery rate was reduced to 60 ± 2% (*n =* 9) with the PtCl_4_ treatment. However, the incubation of HuNoV GII with 20 µg/mL TiO_2_ NPs under UVA reduced the recovery rate to 28 ± 2% (*n =* 9) with or without the PtCl_4_ treatment (*p* < 0.05). In the presence of 2000 µg/mL TiO_2_ NPs and UVA, the recovery rates with or without the PtCl_4_ treatment were reduced to 0.4 ± 4% (*n =* 3) and 0.1 ± 4% (*n =* 3), respectively (*p* < 0.05). Without the TiO_2_ NPs, UVC reduced the HuNoV recovery rates, with or without the addition of PtCl_4_, to 7 ± 3% (*n =* 6) and 41 ± 3% (*n =* 15), respectively (*p* < 0.05). The incubation of HuNoV GII with 20 µg/mL TiO_2_ NPs under UVC similarly reduced the recovery rate, with or without PtCl_4_, to 39 ± 4% (*n =* 3) (*p* < 0.05). The HuNoV recovery rates following its incubation with 2000 µg/mL TiO_2_ NPs and UVC was also reduced to 0.2 ± 4% (*n =* 3) with or without the PtCl_4_ capsid integrity treatment.

The impact of UV and TiO_2_ NPs on MNV was similar to that on HuNoV. Without the UV treatment, the recovery rates of MNV incubated with 20 µg/mL of TiO_2_ NPs were decreased with or without the PtCl_4_ treatment to 21 ± 3% (*n =* 9) and 20 ± 6% (*n =* 9), respectively (*p* < 0.05). They were further reduced in the presence of 2000 µg/mL of TiO_2_ NPs, with or without the PtCl_4_ treatment, to 0.3 ± 0.2 (*n =* 6) and 0 ± 0 (*n =* 6), respectively (*p* < 0.05).

### 3.2. TiO_2_-Coated Plates

The average amount of TiO_2_ recovered from the TiO_2_ NP-coated polystyrene of the 24-well multiple plates was estimated at 892 ± 31 µg/well (469 ± 16 µg/cm^2^) from the ICP measurements, which corresponds to a recovery of 71.7 ± 2.5%. This could result from the solvent-swelling and heating procedures of the material. This approach could lead a proportion of nanoparticles to be integrated into the polystyrene matrix, and thus to be not accessible for dissolution and TiO_2_ quantification. The SEM analysis indicated that TiO_2_ NPs were covering the entire bottom of the wells without apparent aggregates (Figure 2A). Some cracks of around 25 to 54 nm were observed. The AFM scans provided a more precise analysis of the TiO_2_ NP-coated surfaces (Figure 2B). We noticed a low polydispersity and a relatively homogenous distribution on the surface. The average particle size range (5 to 15 nm) was in agreement with the estimated TiO_2_ NP size provided by the supplier.

Since the TiO_2_ NPs were effective against the HuNoV GII without additional UV photoactivation and food is prepared in ambient light, we evaluated the antiviral activity of the TiO_2_ NP-coated polystyrene plates without UV. The average HuNoV and MNV inoculum added to the coated surfaces, as estimated by RT-qPCR, were 2.7 ± 0.5 × 10^6^ gEq and 5.9 ± 0.9 × 10^3^ gEq, respectively. The impact of the TiO_2_ NP-coated surfaces on HuNoV GII and MNV was similar (Figure 3). The recovery of HuNoV GII and MNV from the non-coated wells were 13 ± 3% (*n =* 12) and 16 ± 4% (*n =* 12), the equivalent of 3.5 ± 0.8 × 10^6^ gEq and 9 ± 2 × 10^3^ gEq, respectively. The virus recoveries in the non-coated wells were not altered by the PtCl_4_ treatment. The slight average increase (12 to 19%) observed with both viruses was not statistically significant. Both HuNoV GII and MNV recoveries were significantly reduced, when exposed to the TiO_2_ NP-coated wells, to 5 ± 4% (*n =* 12) and 6 ± 5% (*n =* 12), respectively (*p* < 0.05). Following the PtCl_4_ capsid integrity treatment, the HuNoV GII and MNV recovery yields were further reduced to 1.4 ± 1.4% (*n =* 12) and 1.1 ± 1.6% (*n =* 12), respectively (*p* < 0.05). Compared to the non-coated wells, this was equivalent to an average HuNoV and MNV log reduction of −2.18 ± 0.654 (*n =* 12) and <−2.61 (*n =* 12), respectively.

Long RT-qPCR is another approach that has been employed to estimate the extent of genome degradation (Figure 4). Long-strand RNA RT is more sensitive to PtCl_4_ degradation than shorter RT. The inoculum recovery rates of HuNoV GII and MNV from the non-coated wells, as estimated by long RT-qPCR, were 14 ± 4% (*n =* 9) and 24 ± 3% (*n =* 9), respectively. The HuNoV GII and MNV recovery yields were not modified by the addition of the PtCl_4_ capsid integrity treatment and were estimated at 17 ± 4% (*n =* 9) and 25 ± 8% (*n =* 9), respectively. However, when exposed to the TiO_2_ NP-coated surfaces, both the HuNoV and MNV recovery yields were significantly reduced to 1.9 ± 1.9% (*n =* 9) and 0 ± 0% (*n =* 9), respectively. The addition of the PtCl_4_ treatment did not further reduce the recovery yields of both HuNoV GII and MNV exposed to TiO_2_ NP-coated surfaces, which were 1.3 ± 2.0% (*n =* 9) and 3 ± 4% (*n =* 9), respectively (*p* < 0.05).

## 4. Discussion

The germicidal activity of UV is well known. In this study, UVC alone was active against HuNoV GII, while UVA alone did not decrease HuNoV recovery. UVC is known to be absorbed by nucleic acids, causing the formation of photoproducts such as pyrimidine dimers, and its activity also correlates strongly with the length of the genome [42]. It can also degrade virion capsids. Transmission electron microscopy images have shown that the structure of MNV capsids can be completely disrupted with UVC alone [43]. UV-inactivated viruses could still be detected by RT-qPCR since only a small region of the genome is tested. Moreover, the damage to the capsid might not correlate with damage to the genome [44]. The PtCl_4_ capsid treatment could be used to differentiate between intact and damaged virions by interacting with free nucleic acids [23]. In our study, PtCl_4_ further reduced the recovery rates of UVC-treated HuNoV GII, which is indicative of capsid damage. The group of Park and al. [43], using another capsid integrity treatment based on the monoazide dye PMA, also detected a significant decrease (2 log) in MNV capsid integrity following UVC treatment. On the other hand, in our study, the PtCl4 treatment did not further reduce the viral recovery rates following the TiO_2_ treatment, which could indicate that there was no remaining non encapsidated RNA in solution.

In the current study, the antiviral activity of UVA was not significant. A previous study suggested that increased antiviral activity could be the result of hydroxyl radical generation from a photocatalytic reaction on TiO_2_-coated surfaces [43]. The group of Lee and Ko [29] reported not only that TiO_2_ particles were required for UVA to decrease MNV plaque-forming units (less than one log with 500 mJ/cm^2^) but also that their combination did not impact MNV RT-qPCR recovery rates. The group of Park et al. [45] reported that after 60 min of UVA exposition, TiO_2_ NP-coated glass and surface-fluorinated TiO_2_ NP-coated glass decreased MNV plaque-forming units by 0.53 log and 2.6 log, respectively. They reported a small reduction in their MNV RT-qPCR recovery rates (<1 log) following a 10 min treatment with either UVA alone or with TiO_2_-coated quartz tubes [43]. Still, these authors also reported no reduction in HuNoV GI.1 viral RNA within the experimental time range. Some of these discrepancies suggest that the RNA concentrations, as estimated by RT-qPCR, might not sufficiently reflect the extent of virus degradation, as well as some reactivity variation between the viruses.

Since most reports on the antiviral activity of TiO_2_ are associated with its photocatalytic activity, we did not expect TiO_2_ NP antiviral activity without UV against HuNoV GII to be comparable to the combinations of TiO_2_ NPs and UVA or UVC. Indeed, a previous group reported that no MNV reduction was detected when they were exposed for 4 min to TiO_2_ particles without UV or with up to 1500 mJ/cm^2^ UVA [29,46]. Using a UV-cutoff filter, fluorinated TiO_2_ NP-coated glass was ineffective against the bacteriophage MS2 without UV excitation [45]. The enhanced surface area of TiO_2_ NPs in nanodimensional structures compared to bulk TiO_2_ materials could explain some of the discrepancy in reactivity. Anatase TiO_2_ NPs have been reported to have higher photocatalytic properties than other TiO_2_ crystallites [47]. Nevertheless, others have reported some level of virucidal activity of TiO_2_ NPs when not UV-illuminated. TiO_2_ NPs in the dark or under daylight could decrease influenza titres and degrade this RNA virus’s envelope [48]. Anastase TiO_2_ NPs were also reported to reduce human adenovirus, a non-enveloped DNA virus, under dark and ambient light via sorption and inactivation [49]. Nanostructured TiO_2_ thin films composed of anatase, rutile crystallites, and carbon were recently reported to exhibit antimicrobial activity against a range of pathogens, such as *E. coli*, *S. aureus*, *P. aeruginosa*, and *S. cerevisiae*, without UV activation, although with the UV photocatalyst, the TiO_2_ thin film was more efficient by an order close to one log [50]. Similarly, TiO_2_-coated fluoroplastic mediated the photocatalytic inactivation of astrovirus, rotavirus, and feline calicivirus using visible light (0,18 to 0,88 log reduction) but at a lower extent than with UV light photoactivation (1.75 to 2.78 log reduction) [51]. In the current study, the TiO_2_ NP activity was dose-dependent. The TiO_2_ NP antiviral activity was increased at 2000 µg/mL, while the group of Lee, Zoh, and Ko [46] reported a lower antiviral activity with 1000 µg/mL of TiO_2_ particles. They hypothesized that increasing TiO_2_ particles to this level blocks UV penetration and virus exposure to UV light. Differences in TiO_2_ crystallite structures (Rutile:Anatase/85:15 mix) and preparation, UV intensity, and target viruses might explain the observed variations between the two studies. In our setting, we could not exclude that there could be some interference from high TiO_2_ concentrations that could explain the absence of increased activity associated with TiO_2_ UV photoactivation.

Based on these results, we explored the antiviral activity of TiO_2_ NP-coated surfaces. The SEM and AFM analyses of the TiO_2_ NP-coated polystyrene surface indicated that the selected coating process developed and applied in the present study allows for a relatively homogenous coating. The selected RNA extraction process involved the use of chaotrophic agents to denature proteins, inhibit RNAse, lyse viral capsids, and extract viral RNA. RNA greater than 1000 nucleotides in length have been shown to have low affinity to TiO_2_ materials when using chaotrophic agents at pH 8 [52]. However, the extraction process did not allow us to differentiate some of these mechanisms, including photocatalytic degradation and adsorption, when the lysis is incomplete, for instance, owing to aggregation. A detailed understanding of the virus removal mechanism by TiO_2_ NPs could be important for the development of potential applications [49].

In this study, the TiO_2_ NP-coated plate had a similar level of antiviral activity compared to the TiO_2_ NP suspension against both noroviruses without direct UV photoactivation. However, since the incubation time was 2 h, we could not exclude that some level of the photoactivation process was occurring under ambient light and that reactive oxygen species were generated and altering the viruses. As could be observed with the controls without the coating, some antiviral activity, ~0.9 log reduction, could be associated with the 2 h incubation time and attributed to desiccation as well as binding to the polystyrene surface combined with limited RNA extraction efficiency. Highly hydrophobic plastics such as polystyrene are known to bind viruses. The faeces and virus culture samples tested contained infectious, intact viruses as well as incomplete, non-infectious viruses that could be more susceptible to viral RNA degradation by the remaining enzymes, RNases, or PtCl4 treatment in the tested conditions. Still, the level of HuNoV extracted from the control wells represent more than 3000 times the HID_50_. The remaining antiviral activity, an additional ~1 log reduction from the controls levels, appears to be the result of the TiO_2_ NP coating itself. The TiO_2_ NP coating increases the surface area, and virus binding could explain part of this additional antiviral activity. While some traces of the HuNoV long-strand genome could be recovered, there was no remaining long-strand MNV after the 2 h incubation period on the TiO_2_ NP-coated plates. In terms of the difference in inoculum level, the MNV inoculum was lower by 2.7 log compared to HuNoV, and the assay sensitivity limits could explain, in part, these differences. A higher inoculum concentration could provide a more precise estimation of antiviral activity, as well as experimental conditions, to evaluate HuNoV infectivity reductions in HIE culture systems. However, when there is no or few viral RNA genome remaining, there are not enough viral particles to trigger viral replication and infection. Recently, the HIE has been shown to be a more precise model than PMAxx-based capsid integrity assays to assess HuNoV infectivity following heat or high pressure treatments since viral replication was more susceptible to treatments than its RNA [53]. In this study, there was almost no intact viral RNA remaining, as we observed with the TiO_2_ NP suspension and T_I_O_2_ NP-coated surfaces, indicating that there should only be a low level of remaining infectious virus, if any. Exploring alternative extraction processes, or further analysing virus degradation byproducts, for instance, by mass spectrometry, might also provide additional insights into the antiviral mechanisms of the TiO_2_ NP surfaces [54]. Overall, HuNoV GII were spiked at more than 20,000 times the HID_50_. Compared to the inoculum, the TiO_2_ NP-coated polystyrene resulted in a total signal reduction for HuNoV-intact virus genomes of 98.7 ± 2%.

Several parameters remain to be investigated to better define the potential applications of such TiO_2_ NP-coated polystyrene. These included testing the impact of various incubation times, temperatures, and following incubation in the dark. The target pathogens should also be investigated to compare TiO_2_ NP surface virucidal activity towards enveloped vs. non-enveloped viruses, as well as DNA vs. RNA viruses. For instance, attachment, aggregation, or photoactivation mechanisms could be impacted by the selected pH and control conditions used in this study and be specific to viruses with similar capsid isoelectric points and structures. In addition, organic matter and protein residues may reduce the level of binding as well as the photocatalytic activity, if any, of TiO_2_ NPs by radical scavenging. For instance, the impact of the visible light-catalytic TiO_2_ film on feline calicivirus titres was reduced by one log with the addition of 1 mg/mL bovine serum albumin [51]. In the current study, the TiO_2_ NP-coated surfaces were efficient even when the tested samples contained 1% faeces (~ 50 µg proteins). The food residue’s impact on TiO_2_ NP-coated polystyrene virucidal activity remains to be investigated as well.

Another aspect that remains to be evaluated is the safety of the TiO_2_ NP-coated polystyrene, despite its relative broad use in food products [31]. Part of the concern is the lack of available data on TiO_2_ nanoparticle genotoxicity, which could be difficult to test owing to their low solubility [34,36]. For a coated surface, the migration of the integrated TiO_2_ NPs to the food matrices would also be an important aspect to be verified. For instance, TiO_2_ NPs could migrate from the polylactic/TiO_2_ nanocomposite films to the cheese surface but far below the migration limit of 10 mg/kg defined by the European Food Safety Agency for food contact materials [55].

The potential application of TiO_2_ NPs in food packaging materials and self-sterilizing surfaces is promising [56]. In addition to their antiviral activity and increased UV protection, antimicrobial TiO_2_ NPs could increase the shelf-life of food and inhibit biofilm development. Several groups have already explored TiO_2_ NP applications, mainly as a nanocomposite with chitosan, nanofibers, or polylactic acid. These applications include antiviral TiO_2_ NP-coated cotton fibre, prolonging the storage life of tomato fruits, and antimicrobial cottage cheese packaging [55,57,58]. Since HuNoVs are associated with 58 to 65% of known causes of foodborne illnesses in North America, reducing the food transmission pathway even slightly could have an impact.

## 5. Conclusions

HuNoVs are the leading cause of foodborne illness in several countries. We reported the impact, with and without UV activation, of TiO_2_ NPs on HuNoV GII and MNV. A new process of preparing TiO_2_ NP-coated microplate polystyrene surfaces using a solvent and soft heating was also developed. Under ambient light, these surfaces significantly reduce the level of HuNoV GII and MNV detected. The extent of virus integrity impact was better reflected using the capsid and genomic integrity approaches. Our results support other observations on the antiviral activity of TiO_2_ NP materials. However, the mechanisms involved remain to be investigated. TiO_2_ NP-coated surfaces could have potential practical applications to reduce norovirus transmission associated with contaminated food surfaces. Additional work is required to develop such an application.

## Figures and Tables

**Figure 1 foods-13-01527-f001:**
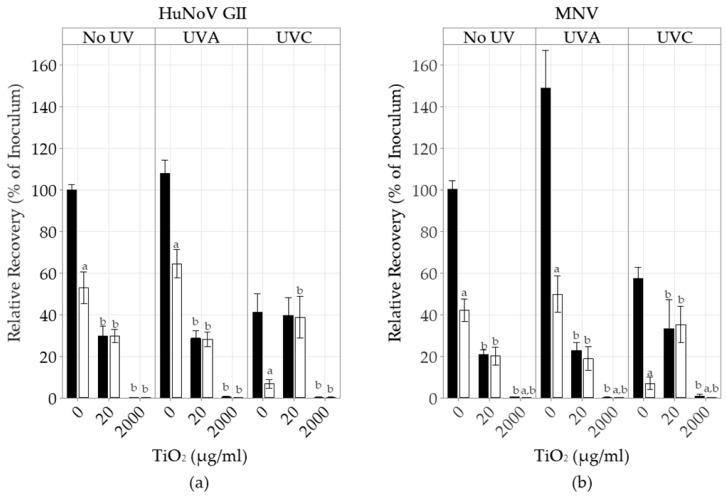
The impact of UV and TiO_2_ NPs on HuNoV GII and MNV recovery with or without the PtCl4 capsid integrity treatment, as detected by short RT-qPCR. HuNoV GII were incubated with different concentrations of TiO_2_ NPs with or without the UV (UVA; UVC) treatments. After this treatment, HuNoV GII (**a**) and MNV (**b**) were either incubated with PBS (*■*) or the capsid integrity agent PtCl_4_ (*□*). The RNA was extracted and detected by short RT-qPCR. The results are expressed as the percentage of HuNoV GII inoculum recovery (mean ± 95% CI). The Friedman test, followed by the post-hoc Conover test (*p* < 0.05), was used to identify differences between the PtCl4 treatments (^a^) or between the TiO_2_ treatments (^b^) and their respective controls.

**Figure 2 foods-13-01527-f002:**
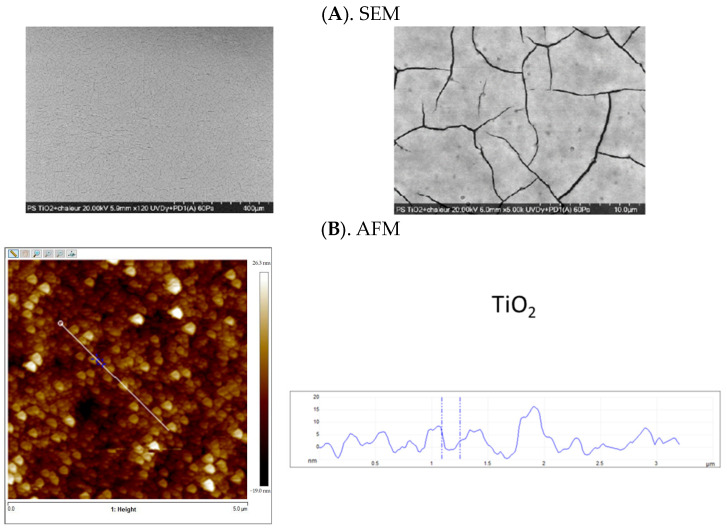
Surface analyses of the TiO_2_ NP-coated polystyrene wells. (**A**) SEM scanning images of the TiO_2_ NP-coated polystyrene plate well bottoms using a scale bar at 400 µM (**left panel**) and at 10 µm (**right panel**). (**B**) AFM scanning image of the TiO_2_ NP-coated polystyrene plate bottom: a 2D image of a typical topography observed (**left panel**) and the height profile of the domains located under the white line of the 2D scan (**right panel**).

**Figure 3 foods-13-01527-f003:**
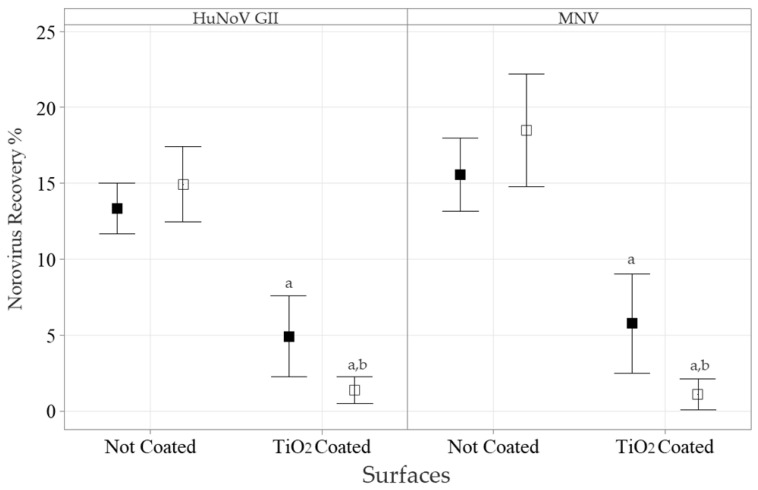
The impact of TiO_2_ NP-coated surface on HuNoV GII and MNV recovery with or without the capsid integrity treatment, as detected by short RT-qPCR. HuNoV GII (**left panel**) and MNV (**right panel**) were incubated for 120 min on TiO_2_ NP-coated polystyrene surfaces. Then, 200 µL of PBS (■) or the PtCl_4_ (□) based capsid integrity treatment was added. The RNA was extracted and detected by short RT-qPCR. The results are expressed as percentages of inoculum recovery (mean ± 95% CI). The Friedman test, followed by the post-hoc Conover test (*p* < 0.05), was used to identify differences between the TiO_2_ NP surface coatings (^a^) or between the PtCl_4_ treatments (^b^) and their respective controls.

**Figure 4 foods-13-01527-f004:**
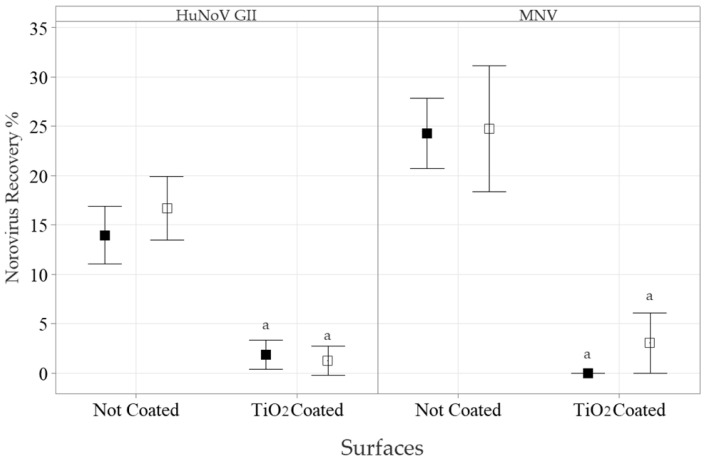
The impact of TiO_2_ NP-coated surfaces on HuNoV GII and MNV recovery with or without the capsid integrity treatment, as detected by long RT-qPCR. HuNoV GII (**left panel**) and MNV (**right panel**) were incubated for 120 min on TiO_2_ NP-coated polystyrene surfaces. Then, 200 µL of PBS (■) or the PtCl_4_ (□) based capsid integrity treatment was added. The RNA was extracted and detected by long RT-qPCR. The results are expressed as percentages of inoculum recovery (mean ± 95% CI). The Friedman test, followed by the post-hoc Conover test (*p* < 0.05), was used to identify differences between the TiO_2_ NP surface coatings (^a^) and their respective controls.

**Table 1 foods-13-01527-t001:** Anatase TiO_2_ NP dispersion characteristics.

Appearance	Crystal	pH	Particle Size nm	TiO_2_ wt %	Solvent wt %	Surface Treating	Na ppm	Al ppm	Cl ppm	TiO_2_ %
Translucent liquid	Anatase	1–4	5–15	≥15.3	Water~85	No	≤98	≤95	≤93	99.9

## Data Availability

The original contributions presented in the study are included in the article, further inquiries can be directed to the corresponding author.

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
