# Peer review of "Impact of Nanoparticle-Based TiO2 Surfaces on Norovirus Capsids and Genome Integrity"

_foods, 2024, doi:10.3390/foods13101527_

Round 1
Reviewer 1 Report
Comments and Suggestions for Authors
In the manuscript, the antiviral activity of titanium dioxide nanoparticles (TiO2 NP) against MNV and HuNoV, human and murine noroviruses, is examined. Significant antiviral effects on contaminated surfaces are revealed by investigating both UV-activated and non-activated TiO2 NP. Beyond stressing safety concerns and future research directions, the study proposes TiO2 NP-coated surfaces as viable instruments for stopping virus transmission in food packaging. Readers will find the study intriguing, and it is worth reading. But this work still needs to be improved in the following ways:
1. Further elucidation is required in the manuscript regarding the ideal circumstances for TiO2 NP activity and its efficacy against various viral species.
2. The specificity of TiO2 NP activity should be discussed by the authors, especially in terms of differentiating between DNA and RNA viruses as well as enveloped and non-enveloped viruses.
3. What effect does organic matter have on the virucidal activity of TiO2 NP, such as food residues or protein residues in excrement?
4. The conclusion should give a succinct overview of the key findings, their importance within the context of the research field, and the study's limitations.
Reviewer 2 Report
Comments and Suggestions for Authors
In this article, the authors reported the TiO2 coated surface could reduce the recovery rates of intact HuNoV. I think it could be considered to be published in this journal after major revision.
1. The experimental results are too little, which could not support all the results the authors proposed in this article.
2. Why the authors did not test the recovery rates of intact HuNoV treated by TiO2 coated surface in the dark? It is very important because the TiO2 might also be activated by ambient light.
3. The authors should prove if some HuNoV were adsorbed on the TiO2 coating surface, because the adsorption could also lead to an intuitive decrease of recovery rates.
4. The recovery rates of HuNoV GII and MNV on the polystyrene surfaces without TiO2 are already very low. The authors should explain it. The TiO2 coating could increase the contact area, which means that the adsorption of HuNoV GII and MNV on the TiO2 might be the real reason to reduce the recovery rates.
5. There are many grammatical and writing mistakes in this version.
Comments on the Quality of English Language
Need to improve English.
Reviewer 3 Report
Comments and Suggestions for Authors
The manuscript entitled “Impact of Nanoparticle Based TiO2 Surface on Norovirus Capsid and Genome Integrity” describes the impact of a 2000 µg/ml TiO2 NP with or without UV activation on HuNoV GII.4 and murine norovirus on their recovery rates, and also describe a new TiO2 NP coating process on polystyrene surface. The work is relevance due to the impact of norovirus on public health due to acute gastroenteritis outbreaks, and to biological characteristic of these agents that contribute to the problem. I have made a few suggestions and comments to improve the manuscript, so it is considered for publication.
INTRODUCTION
Page 1, lines 37- 38: I suggest that the authors modify the sentence accordingly and include the manuscript by Wang et al., 2023 (Wang et al. BMC Infectious Diseases (2023) 23:595 https://doi.org/10.1186/s12879-023-08519-y) as a reference after the sentence “HuNoV are shed in the vomit and faeces of people who are infected including by asymptomatic carriers”.
Page 1, line 38: I suggest replacing the fragment “HuNoV can survive” to “HuNoV can resist inactivation” ou “ remains stable”.
Page 2, between lines 51-52: I suggest the authors mention the existence of the organoid model as a tool to evaluate some of the norovirus genotypes infectivity in vitro, even though it is not readily available in all lab, and include a reference from the Estes lab.
I also suggest that a sentence about the stability of viruses on surfaces and environments is included in the Introduction, considering that the stability also influenced by the type of surface, temperature and the medium (environment) in which the viruses are in (example: if there is organic matter the viruses may be more resistant to inactivation).
METODOLOGY
I believe the authors used an appropriate methodology; however, virus infectivity was not evaluated, only a decrease in genomic copy numbers and damage to the capsid and genome, this should be very clear throughout the manuscript.
DISCUSSION
As limitations, the discussion should address the fact that viral infectivity could not be accessed. This could be evaluated by using the organoid model.
GENERAL COMMENTS AND SUGGESTIONS
One suggestion would be the use of Western blotting (and similar methods) to assess the integrity of specific capsid domains (P and S domains) post-treatment, to see whether the structural degradation of the capsid is domain target or if it affects quaternary structure of the capsid monomers. By doing so one could have new insights on how to detect proteolytic degradation or aggregation of capsid proteins uponTiO2 NP treatment. Or is there any Ti byproduct that could be detected and demonstrated as a product of capsid/genome degradation?
Also, is there an effect of different concentrations of TiO2 particle and varying exposure times in the disruption of capsid integrity and antiviral activity? Is activity only observed after a determined time of exposure?
Testing the infectivity of treated viruses in relevant animal models or organoid systems can help establish whether structural/genomic damage to the capsids correlates with a loss of infectivity and efficient delivery of the genome intracellularly.
Would it be possible to use any microscopy to visually assess the physical “damage” in the capsid structure caused by nanoparticle treatment? This could be backed up by the results of the western blot-like assay.
Also, how long does the nanoparticle coating would last ideally? Is there an limit amount of virus that could be deactivated in a given area (for example cm2 of coated material)?
Round 2
Reviewer 1 Report
Comments and Suggestions for Authors
The authors have successfully addressed all the raised issues in the manuscript. I recommend acceptance of the manuscript for publication in its present form.